# Photodynamic Therapy by Mean of 5-Aminolevulinic Acid for the Management of Periodontitis and Peri-Implantitis: A Retrospective Analysis of 20 Patients

**DOI:** 10.3390/antibiotics11091267

**Published:** 2022-09-18

**Authors:** Roberto Rossi, Lorena Rispoli, Michele Antonio Lopez, Andrea Netti, Morena Petrini, Adriano Piattelli

**Affiliations:** 1Private Practice, 16121 Genova, Italy; 2Department of Periodontology, Humanitas Dental Center, Humanitas Research Hospital, Rozzano, 20089 Milano, Italy; 3Department of Head and Neck and Sensory Organs, Division of Oral Surgery and Implantology, Fondazione Policlinico Universitario A. Gemelli IRCCS-Università Cattolica Sacro Cuore, 00168 Rome, Italy; 4Department of Medical, Oral and Biotechnological Sciences, University of Chieti-Pescara, 66013 Chieti, Italy; 5School of Dentistry, Saint Camillus International University of Health and Medical Sciences, 00131 Rome, Italy

**Keywords:** periodontal disease, peri-implantitis, aminolevulinic acid, photodynamic therapy

## Abstract

Periodontitis and peri-implantitis are common in the population worldwide. Periodontal diseases affect approximately 50% of adults, while mucositis affects 80% of patients with implants, turning into peri-implantitis at a rate varying from 28 to 58%. If standardized treatments for all degrees and variety of periodontal diseases are known and codified, a consensus on the treatment of peri-implantitis still has to be found. Photodynamic therapy (PDT) has been used successfully in the medical field and was recently introduced as supportive therapy in dentistry. This paper reviews the results on 20 patients, 10 affected by periodontal disease (grades II to III) and 10 by peri-implantitis. Application of 5% 5-aminolevulinic acid gel (ALAD), as a support of causal therapy, in periodontal pockets and areas of peri-implantitis favored the maintenance of severely compromised teeth and significantly improved compromised implant conditions. Between baseline and 6 months, all teeth and implants remained functional. All patients confirmed that the scaling and root planning (SRP)+ALAD-PDT was not painful, and all perceived a benefit after the treatment at all timing points. For periodontal patients, a significant decrease in PPD after 3 (*p* < 0.001) and 6 months after SRP+ALAD-PDT respect baseline values were observed. For the implant patients, the SRP+ALAD-PDT was correlated to a decrease in PPD and BOP, and a slight increase in the number of exposed threads. However, the results were statistically significant only for PPD (*p* < 0.001).

## 1. Introduction

The Global Burden of Disease Study 2019 estimated that oral diseases affect close to 3.5 billion people worldwide, and periodontal diseases affect about 50% of living human beings [1]. Periodontal disease is part of the chronic multi-factorial inflammatory diseases; it is related to the composition of the dental plaque biofilm and is characterized by a progressive loss of supporting structures [2,3].

In 2018, a joint venture between the European Federation of Periodontology (EFP) and the American Academy of Periodontology (AAP) proposed a new classification for periodontal and peri-implant diseases [4]. The mildest form is gingivitis, a reversible situation caused by biofilm accumulation. Periodontitis, the more aggressive disease, was divided into four stages on the basis of the severity of the lesions and complexity of its management: initial (stage I), moderate (stage II), and severe periodontitis with potential for additional tooth loss (stage III), and with potential for loss of dentition (stage IV). The risk of progression was assessed as slow (grade A), moderate (grade B), and rapid (grade C).

Peri-implantitis occurs similarly to gingivitis-periodontitis; it starts as mucositis and evolves into peri-implantitis. Data suggest that its progression is faster than the one observed in periodontitis; 80% of patients with implants have mucositis, and 28–58% of cases evolve into peri-implantitis [5]. Assess periodontal disease is common practice; scaling, root planning, and good oral habits and motivation help prevent its progression [6].

Non-surgical therapy, when combined with proper home care, can help patients maintain the status quo and prevent the development and or progression of periodontal disease [7]. On the other hand, non-surgical therapy has demonstrated several limitations derived from the local anatomy (tooth shape, furcation defects, bone defects, inaccessible surfaces) [8]. This is one of the reasons why initial therapy has often been improved by adding adjunctive means such as air polishing with erythritol, laser, and antibiotics [9,10,11].

Mechanical debridement is often considered sufficient to decontaminate implants; however, it must always be accompanied by mechanical and chemical home oral hygiene procedures. The goal of the procedure is to eliminate supra- and subgingival biofilm using hand instruments and ultrasonic. Plastic-coated or carbon-fiber tips were introduced to perform mechanical debridement, avoiding scratches on the implant surfaces. The remaining plaque and calculus amount should be removed using titanium, carbon fiber, or plastic curettes [12].

### Photodynamic Therapy

Photodynamic therapy (PDT) has surfaced as a very effective supporting treatment of both periodontitis and peri-implantitis; it involves the use of a photosensitizer applied to the gums and adjacent to the teeth or implants. The substance is then activated with the light of a specific wavelength [13,14,15,16]. The light activation causes changes from the ground singlet state to the excited triplet state, which, reacting with environmental oxygen, can produce highly reactive singlet oxygen and toxic reactive oxygen species (ROS) [17] in the dye molecule, which oxidizes and destroys bacterial cells [18].

In 2020, Lopez et al. conducted a systematic review on the efficacy of PDT in the treatment of periodontitis and peri-implantitis. Out of 90 studies published, 7 were included, 4 addressing the need for PDT in peri-implantitis and 3 for periodontitis. All the studies considered reported the outcomes of PDT on the multi-bacterial species found in those pockets; all studies demonstrated a significant reduction of the bacterial load, with an average reduction of 99.3%. Clinical parameters also improved with probing pocket depth (PPD) reduction of 1.01 mm, bleeding on probing (BOP) of 50%, and relative clinical attachment level (RCAL) of 1.19 mm. These results were considered preliminary but encouraging [19].

In recent years, research has identified several photosensitizing substances, including methylene blue, toluidine blue, and malachite green/indocyanine green [20]. Radunovic et al. (2020) published a paper describing the application of a novel gel containing 5-aminolevulinic acid (ALAD) and 630 nm red light (ALAD-PDT) to treat cases of periodontitis and peri-implantitis. The test demonstrated in vitro its effectiveness against different bacterium types involved in the infections of the oral cavity. A preliminary experiment tested the effectiveness of the gel after one hour of incubation vs. *Enterococcus fecalis*; in a second stage of the experiment, ALAD was tested, with and without irradiation, on *Staphylococcus aureus, Enterococcus fecalis, Escherichia coli, Veillonella parvula*, and *Porphiromonas gingivalis* [21]. The efficacy of the gel, in different concentrations, after 25 min of incubation was tested with and without led light illumination. This research produced the most effective result by 25 min of ALAD application followed by 5 min of light irradiation [21]. The same protocol was tested on implant surfaces and proved effective against Pseudomonas aeruginosa [22]. In vitro tests also demonstrated a significant reduction of *Streptococcus oralis* on machined and double-etched implant surfaces [23].

This study aims to evaluate the clinical effects of 5% 5-aminolevulinic acid gel (Aladent^®^, Alpha Instruments, Milan, Italy) as an adjunct to non-surgical therapy in patients treated for periodontitis and peri-implantitis.

## 2. Results

### 2.1. Results on Periodontal Sites

A total of 10 periodontal sites were evaluated at t0 and after 3 and 6 months after SRP+ALAD-PDT. The 10 patients were 5 males, and 5 females, and the average age was 61 ± 8. All patients confirmed that the SRP+ALAD-PDT was not painful, and all perceived a benefit after the treatment at all timing points. The Levene test confirmed the homogeneity among the variables, and the ANOVA found significant differences at the three timing points concerning PPD (*p* < 0.001), BOP (*p* = 0.001), and MOB (*p* = 0.034) (Figure 1). A significant decrease in the three parameters was recorded after the treatment SRP+ALAD-PDT, and results were maintained until 6 months of follow-up. The average values ± standard deviation is shown in Table 1.

The LSD test showed a significant decrease in PPD after 3 (*p* < 0.001) and 6 months after SRP+ALAD-PDT with respect to baseline values. Moreover, BOP showed a significant decrease at 3 (*p* = 0.004) and 6 months (*p* = 0.001), compared with T0. The same trend was also shown by MOB with *p*-values of *p* = 0.023 at 3 and 6 months. No significant differences were found between REC and the benefits perceived by the patients. The perception of gained benefits remained constant throughout the six-month follow-up.

### 2.2. Results on Peri-Implant Sites

A total of 10 peri-implantitis sites were evaluated at T0 and after 3 months from SRP+ALAD-PDT. The 10 patients were 6 males and 4 females, and the average age was 58 ± 14. All patients confirmed that the SRP+ALAD-PDT was not painful, and all perceived a benefit after the treatment at all timing points. The Levene test confirmed the homogeneity among the variables, and the t-test was performed. The average values ± standard deviation of clinical parameters are shown in Table 2.

The SRP+ALAD-PDT was correlated to a decrease in PPD and BOP, as well as a slight increase in the number of exposed threads. However, the results were statistically significant only for PPD (*p* < 0.001).

## 3. Discussion

Periodontitis and peri-implantitis are widespread infectious diseases in adults, and their management mainly aims to remove supra- and subgingival deposits, thus eliminating pathogenic bacteria. SRP is an indispensable part of the initial nonspecific treatment protocol. However, the presence of inaccessible areas and resistant microorganisms has led to the continued search for additional therapies, such as a-PDT, before moving to surgery or when surgery is not possible [24].

The data obtained in our study showed that the ALAD-PDT procedure, in combination with SRP, showed promising results in the treatment of periodontitis, with statistically significant clinical improvements in PPD, BOP, and MOB. There were also clinical improvements (PPD, BOP) in treated peri-implantitis cases, statistically significant only for PPD. CAL was not reported directly as a parameter because it is an indirect parameter that can be derived from PPD and REC values. Increased REC at sites with periodontitis and exposed threads at implant sites is related to decreased tissue edema associated with reduced inflammation. The difference is slight and not statistically significant, although it may result in a greater aesthetic deficit. Reduced MOB at sites with periodontitis is a sign of reduced tissue edema. Although MOB is not a recommended clinical sign for assessing the state of periodontal health or disease, because it is possible to have mobility with a reduced but healthy periodontium [25], it was a data point that, in the investigations performed, could give us information. The perceived benefits obtained remained constant during the six-month follow-up for all 20 patients, confirming the excellent tolerability and non-invasiveness of the treatment. Improvements in marginal bone loss were also observed in some cases, but the nonhomogeneity of the radiographic material did not allow the comparison of different cases over time to be detected (Figure A1).

In an in vivo study by Lauritano et al. [26], Aladent^®^ was used to treat periodontitis sites and compared with SRP treatment alone. Aladent^®^ showed a significant reduction in total bacterial load (75%) compared with sites treated with SRP alone, in agreement with previous in vitro studies [21,23]. In the study by Lauritano et al., the mean CAL/PD and percentage of BOP, recorded after SRP + Aladent^®^ therapy, showed statistically significant reduction (CAL from 4.65 to 2.42, PD from 5.14 to 2.87 mm, 90% of sites without bleeding) compared with those obtained after treatment with SRP alone (CAL from 4.73 to 4.08 mm, PD from 5.24 to 4.73 mm, 70% of sites without bleeding) [26].

Aladent^®^ contains aminolaevulinic acid, which is the precursor of protoporphyrin IX (PpIX) in EME biosynthesis and is normally synthesized in mitochondria through the condensation reaction between glycine and succinyl-CoA [27]. The scientific rationale for ALAD-PDT is the induction of protoporphyrin IX (PpIX) in certain bacterial species that showed a peak after 45 min of incubation with ALAD [28]. PpIX acts as a specific photosensitizer that, after irradiation with red light (wavelength 630 nm), induces ROS production, directly and indirectly causing cytotoxic effects in high-metabolism cells such as cancer cells and microbes. ROS penetrate cell membranes, inducing oxidation of amino acids and membrane lipids, agglutination of proteins, and oxidative damage to nuclear acids [29]. Moreover, it has been shown that the use of a red led, radiating ALAD, allows the antimicrobial effect to be maintained longer [28].

In addition, the new Aladent^®^ contains the preservatives potassium sorbate and sodium benzoate, which enhance the bactericidal effect [30]. Several studies have analyzed the effects of PDT in treating periodontitis and peri-implantitis.

### 3.1. PDT in Periodontitis

While some authors report statistically significant improvements in PPD reduction and/or clinical attachment level (CAL) following SRP + PDT compared to SRP alone [31,32], others found no statistically significant differences in these parameters between groups [33,34], as confirmed by a recent meta-analysis of Salvi et al. that did not identify a statistically significant difference for PPD reduction in favor of PDT (weighted MD = 0.35 mm, *p* = 0.08) [35]. During periodontal maintenance therapy, some authors have reported statistically significant clinical improvements in terms of reduction of BOP and PPD following SRP + PDT compared with SRP alone [36]. These findings support those of a recent systematic review and meta-analysis, whose analyses support further clinical improvement with PDT in the treatment of residual periodontal pockets in maintenance therapy [37]. However, Siva et al., in their study, showed no further clinical improvements in PDT compared with SRP alone [38]. The use of PDT has also been studied in patients with diabetes mellitus with more advanced forms of periodontitis (stage III, grade C), reporting further clinical improvements in deep pockets compared with SRP alone [39].

### 3.2. PDT in Peri-Implantitis

Previous studies have demonstrated the effectiveness of PDT therapy in reducing bacterial load in vitro [14] and in vivo [40]. A study by Deppe et al. [41] demonstrated its effectiveness in preventing further bone loss over 6 months on implants with moderate bone loss. In other RCTs in which sites of incipient peri-implantitis (PD 4–6 mm, positive BOP, and radiographic bone loss ≥2 mm) were treated, PDT was effective in reducing mucosal inflammation but no more so than the local application of antibiotics. 

The ALAD-PDT protocol is another aid to lowering bacterial load without resorting to antibiotics, which is questionable and contributes to antibiotic resistance, a severe global health problem [42].

Limitations of our study include the fact that the retrospective setting does not allow us to exclude possible procedural bias, as well as the lack of a control group with SRP treatment alone or with alternative PDT methods and photosensitive molecules. This is because routinely treated patients with clinical parameters outside the normal range underwent surgical therapies. A further limitation concerns the duration of follow-up: it would be interesting to evaluate long-term data as part of a prospective study, considering even repeated treatments. In addition, it would be clinically meaningful to distinguish deeper from moderately deep probing sites. Moreira et al. demonstrated the efficacy of PDT when PPD was ≥7 mm achieving statistically significant clinical improvements in PPD reduction [43].

Among the study’s strengths are that a single experienced practitioner performed the treatments, and the data were evaluated and analyzed separately by other authors. In addition, some patients were excluded from the data set (e.g., systemic diseases, antibiotics taken for other conditions, multi-rooted teeth), as reported in the Materials and Methods, which would have made the analyzed sample more heterogeneous. Furthermore, our study analyzed the clinical effects of a single treatment with Aladent^®^, even though the product was created for repeated use, as other PDT protocols also require.

The promising results of this retrospective evaluation will need to be confirmed with future investigations by establishing a prospective RCT, with a larger sample, that can confer statistical strength to the differences. However, the positive effects observed in this small group may suggest that ALAD could be a potentially beneficial biomaterial in treating periodontitis and peri-implantitis.

## 4. Materials and Methods

This study is a pilot, retrospective evaluation of a case series extrapolated from a database of patients treated for periodontitis and peri-implantitis between January 2021 and March 2022 in a private clinic in Genoa, Italy. To make the sample more homogeneous, patients’ medical history was analyzed to exclude those with systemic diseases (e.g., diabetes, autoimmune diseases, hormonal disorders), history of taking antibiotics immediately before or in the months of post-treatment observation, pregnancy, or smoking habit >10 cigarettes per day.

Twenty subjects were identified with 20 sites, 10 with grade II–III periodontitis and 10 with peri-implantitis, according to the most up-to-date diagnostic criteria [4]. Among the periodontal sites, only treated teeth with a single root were included to have a homogeneous sample. The treatment was part of causal periodontal therapy or post-treatment maintenance. All patients underwent a follow-up visit and oral hygiene where some residual probing pockets ≥ 5mm around teeth or ≥6 mm around implants were observed. Two weeks after the recall visit, the sites were treated with the application of Aladent^®^ protocol. All treated patients reported a full-mouth plaque score (FMPS) ≤ 20 and a full-mouth bleeding score (FMBS) ≤ 20. The report of this study complies with the STROBE statement.

### 4.1. Clinical Procedure

Treatments were performed by the same experienced clinician (R.R.) using magnification and the same procedural protocol. ALAD gel was inserted into each pocket with a dedicated small tip applied to the dispenser syringe, and then the area was protected with a light-cured gingival barrier (Opal Dam^®^, Ultradent, Milan, Italy) and left in place for 60 min. After removal of the gingival barrier, reaction was checked with the dedicated FL-02 flashlight, observing the presence of the photosensitizer (PpIX). The FL-02 is a portable LED emitter tuned to the Soret band of PpIX (395 ± 10 nm). This allows for the viewing of the specific red/purple fluorescence (635 nm) of PpIX without interference from other fluorophores. Due to the low power of the emitter, observation should be carried out in a dark environment. Thus, the site was illuminated for 7 min with a TL-01 LED light of 630 Nm and 40 joules per sq/cm (high-power LED light source, Alpha Strumenti, Milan, Italy) (Figure 2).

The illuminator tip was placed perpendicularly and close to the surface of the gingival mucosa. After treatment, all patients were asked to follow appropriate oral hygiene maneuvers.

Probing pocket depth (PPD), bleeding on probing (BOP), gingival recession (REC), and mobility (for the teeth) (MOB) were recorded at baseline and at 3 and 6 months. PPD and BOP were recorded using a probe (Teflon for implants). PPD, measured in millimeters from the mucosal margin to which a probe penetrates into the pocket and evaluated at six sites around each implant or tooth (buccal, palatal/lingual, mesial, distal, mid); REC, measured in millimeters from the implant neck or tooth cemento-enamel junction (CEJ) to the mucosal margin and evaluated at six sites around each implant or tooth (buccal, palatal/lingual, mesial, distal, mid); BOP, bleeding induced by gentle manipulation of a probe to gingival sulcus tissue, recorded at six sites around the implant or around teeth. Miller’s index was used to assess MOB: the tooth is held and moved between the metal handles of two instruments and in the buccolingual direction; the distance moved is visually estimated by the person performing the examination. Mobility is then classified into grades 0–3: mobility grade 0—horizontal mobility of the crown less than 0.2 mm; grade 1—horizontal crown mobility between 0.2 and 1 mm; grade 2—horizontal mobility of the crown between 1 and 2 mm; grade 3—tooth displacement in the vertical (apico-coronal) direction added to grade 2. All patients filled up a visual analogue scale (VAS) from 0 to 10 points requesting at 1 week after treatment and at 3 and 6 months, the benefits received by therapy, as routinely reported after each treatment.

### 4.2. Statistical Analysis

Statistical analysis was performed using SPSS for Windows version 21 (IBM SPSS Inc., Chicago, IL, USA). The Levene test permitted the evaluation of the homogeneity of the groups. The analysis of variance (ANOVA) and Fisher’s least significant difference (LSD) test were used to compare the clinical parameters on periodontal sites analyzed at T^0^, 3 months, and 6 months. The t-test was used for viability assay on peri-implantitis sites compared at T^0^ vs. 3 months. A *p*-value < 0.05 was considered significant.

## 5. Conclusions

The results showed that in periodontal patients, all sites evaluated had a beneficial effect on PPD, BOP, and MOB with statistically significant differences between baseline and at 3 and 6 months after. No significant differences were found for REC, and the perception of gained benefits remained constant throughout the six-month follow-up. For the peri-implantitis sites, all patients confirmed that the SRP+ALAD-PDT was not painful, and all perceived a benefit after the treatment at all timing points. The SRP+ALAD-PDT was correlated with a decrease in PPD and BOP, and a slight increase in the number of exposed threads. However, the results were statistically significant only for PPD.

## Figures and Tables

**Figure 1 antibiotics-11-01267-f001:**
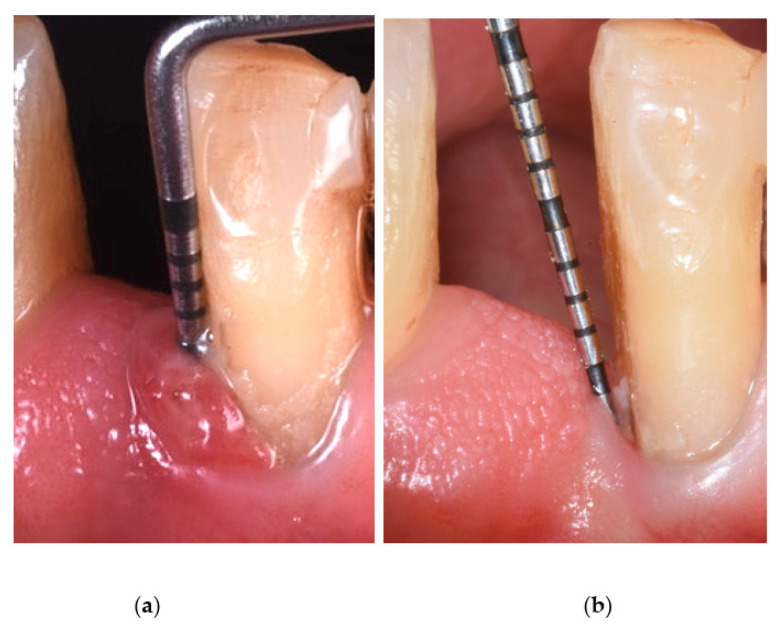
Probing pocket depth before ALAD-PDT treatment (**a**) and after ALAD-PDT treatment (**b**).

**Figure 2 antibiotics-11-01267-f002:**
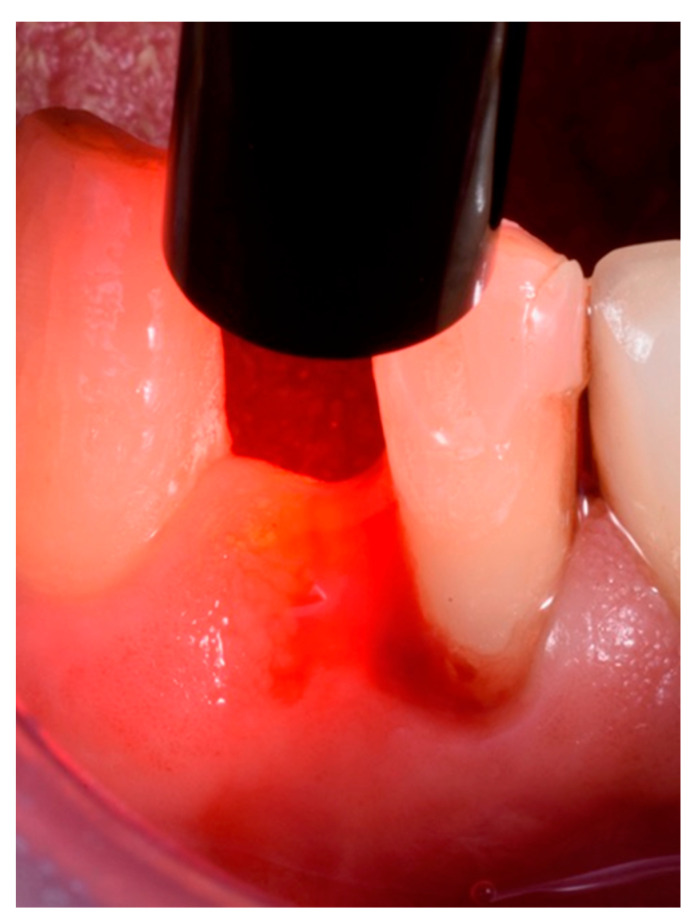
TL-01 LED red light applied to the treated site into which the Aladent^®^ gel was inserted.

**Table 1 antibiotics-11-01267-t001:** Clinical parameters were measured at baseline (T^0^), after 3 months, and after 6 months in periodontitis sites treated with Aladent^®^.

Clinical Parameters	Timing	Average Value	Standard Deviation
**PPD**	T^0^	7.00	1.89
	3 m	3.10	0.74
	6 m	3.00	3.00
**BOP**	T^0^	0.60	0.52
	3 m	0.10	0.32
	6 m	0.00	0.00
**REC**	T^0^	1.50	1.78
	3 m	1.40	1.35
	6 m	1.30	1.34
**MOB**	T^0^	0.30	0.48
	3 m	0.00	0.00
	6 m	0.00	0.00
**PATIENT-** **REPORTED** **BENEFITS**	1 w	9.40	0.70
	3 m	9.00	0.82
	6 m	8.80	0.63

PPD: probing pocket depth; BOP: bleeding on probing; REC: recession; MOB: mobility; m: months; w: week.

**Table 2 antibiotics-11-01267-t002:** Clinical parameters were measured at baseline (T^0^), after 3 months, and after 6 months in peri-implantitis sites treated with Aladent^®^.

Clinical Parameters	Timing	Average Value	Standard Deviation
**PPD**	T^0^	5.60	0.84
	3 m	3.00	1.05
	6 m	3.20	1.63
**BOP**	T^0^	0.60	0.52
	3 m	0.30	0.48
	6 m	0.30	0.63
**EXPOSED THREADS**	T^0^	0.40	0.70
	3 m	0.60	1.07
	6 m	0.55	1.24
**PATIENT-REPORTED** **BENEFITS**	1 w	9.00	0.82
	3 m	8.70	0.75
	6 m	8.60	0.90

PPD: probing pocket depth; BOP: bleeding on probing; m: months; w: week.

## Data Availability

The data presented in this study are available on request from the corresponding author. The data are not publicly available because this is not a clinical trial registered in the public registry of studies, but data from privately treated patients were retrospectively analyzed.

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
