# Peer review of "Photodynamic Therapy by Mean of 5-Aminolevulinic Acid for the Management of Periodontitis and Peri-Implantitis: A Retrospective Analysis of 20 Patients"

_antibiotics, 2022, doi:10.3390/antibiotics11091267_

Round 1
Reviewer 1 Report
Dear Authors,
Congratulations on your research topic.
The pressing need of alternative, minimal invasive approaches to support peri-implantitis treatment has nicely been proven in your introduction.
The identified study limitations mentioned by the authors are relevant and at the same time a call to action for established research institutions.
Your case study deserves publication which I hereby recommend.
Best regards
Author Response
Response Reviewer 1: Dear Reviewer,
Thank you for your work in revising our manuscript. As made explicit in your comments, in the discussion we reported the limitations of our study, following the STROBE guidelines, so that we could set up a study to continue the analysis done, improving the limitations.
We hope that the revised manuscript is suitable for publication.
Best Regards
Reviewer 2 Report
Authors presented interesting study about using photodynamic therapy with 5-ALA in patients suffer from periodontitis and peri-implantitis, which are idespread infectious diseases in adults. Photodynamic therapy has surfaced as a very effective supporting treatment of both periodontitis and peri-implantitis; it is an alternative to antibiotic therapy and other invasive treatments. Authors reveale, that all patients confirmed that the scaling and root planning (SRP)+ALAD-PDT was not painful, and all perceived a benefit after the treatment at all timing points. For periodontal patients, authors described a significant decrease of PPD after 3 (p<0.001) and 6 months after SRP+ALAD-PDT respect baseline values. For the implant patients, the SRP+ALAD PDT was correlated to a decrease in PPD, BOP and a slight increase in the number of exposed threads.
Comments:
1. Please provide a detailed description of the FL-02 flashlight device used to assess the presence of PpIX
2. The results were statistically significant only for PPD, it would be worthwhile to conduct research on more patients.
Author Response
Response Reviewer 2: Dear Reviewer,
Thank you for your work in revising our manuscript. Your comments have allowed us to better clarify the suggested aspects, improving the quality of the manuscript. As the corresponding author, I have provided a point-by-point response.
Point 1: Please provide a detailed description of the FL-02 flashlight device used to assess the presence of PpIX.
Response 1: As suggested, we have added additional technical details about the device, which improve the clarity of the procedure. The text was modified, and we added (269-273): “The FL-02 is a portable LED emitter tuned to the Soret band of PpIX (395±10nm). This allows viewing the specific red fluorescence (635 nm) of PpIX without interference from other fluorophores. Due to the low power of the emitter, observation should be carried out in a dark environment.”
Point 2: The results were statistically significant only for PPD, it would be worthwhile to conduct research on more patients.
Response 2: After this pilot evaluation we would like to organize an RCT and, as suggested, increase the sample size so that any differences can have statistical power. As suggested, in the text we added (lines 240-241): “with a larger sample that can confer statistical strength to the differences”.
We hope that the revised manuscript is now suitable for publication.
Best Regards